**Data Availability Statement:** Data is available via Zenodo at 10.5281/zenodo.8367959.

**Funding:** Funding for this project was provided by the Royal Netherlands Embassy as part of the

# Adapting the informed push model to the last mile of the contraceptive supply chain in South Kivu in the Democratic Republic of Congo

Katherine H. LaNasa[1]*, Michel Yalaza[2], Felix Hitayezu[3], Frank Roijmans[4]

**1** Department of Health Policy and Management, School of Public Health and Tropical Medicine, Tulane University, New Orleans, Louisiana, United States of America, **2** i+solutions, Kinshasa, DRC, **3** i+solutions, Kigali, Rwanda, **4** i+solutions, Woerden, Netherlands

These authors contributed equally to this work.

* kschulze@tulane.edu

## Abstract

In the Democratic Republic of Congo (DRC), contraceptive security is one of the largest barriers to improving access to family planning. This article presents findings from a pilot study that adapted an informed push distribution model to the last mile of the contraceptive supply chain, between the health zone pharmacy and health facilities, in the eastern region of DRC. The intervention consisted of three changes in the supply chain: organization of more efficient transportation routes, in-depth involvement of the community in supply chain management and bundling of contraceptives with other essential medications for efficient delivery. The intervention was implemented from October 2017 to October 2018 in the Katana health zone of the South Kivu province. Although there was not a statistically significant difference in availability of contraceptives during the pilot study period, there were sharp declines in the mean length of stockouts at health facilities and the monthly transportation costs of delivering contraceptives. Overall, the pilot study demonstrated it is feasible to adapt the IPM to a new location with complex geographical, political and socioeconomical influences. Future studies will be required to evaluate whether the adapted informed push model is more effective than the existing pull supply chain system for contraceptives in the DRC.

## Introduction

The Democratic Republic of Congo (DRC) ranks 179th out of 186 countries on human development indicators according to the United Nations [1], reflecting a low life expectancy, educational index and standard of living. The DRC has the third-largest population (86,790,000) [2] and third-highest fertility rate in Sub-Saharan Africa (SSA) [3]. Although the DRC government has shown support for family planning (FP) by establishing the first national strategic plan for family planning in 2014 [4], and revised in 2022 [5], the modern contraceptive prevalence rate (mCPR) among all women has remained low (16.3%), while nearly a third of women (31.8%) have an unmet need for FP [6].

Commodities, Chain and Care Project, led by
UNFPA DRC. The supply chain component of the
project for this study was executed by i+Solutions.
The funders had no role in study design, data
collection and analysis, decision to publish, or
preparation of the manuscript.

**Competing interests:** The authors have declared
no competing interests exist.

Increasing women's access to contraception is a key component to addressing unmet need for FP and increasing mCPR [7,8]. However, ensuring reliable access to contraceptive commodities requires a well-functioning supply chain that can deliver the right type of contraceptives, in the right amount, at the right time to service providers [9,10]. When the contraceptive supply chain does not function properly, stock levels at health facilities are suboptimal, including high rates of stockouts that persist causing unmet need for family planning to increase as women's contraceptive choices are limited due to lack of availability [7,11,12].

There are two common supply chain systems used for commodity distribution, push (allocation) and pull (requisition) [13]. Both supply chain systems require effective and efficient management [14]; the main difference between the two systems relates to who is making the resupply decisions. In a push system, a higher-level authority, such as health zone manager, calculates the quantity of commodities to push down the chain [15]. In a pull system, lower-level health facilities order commodities as needed, thereby pulling the supplies through the chain [15].

Building a strong and efficient supply chain is especially challenging in low- and middle-income countries (LMICs) where resources are limited. Typically, the final link of the supply chain between the health zone (HZ) warehouse and service provider, commonly referred to as the last mile, is the weakest component of the system [16]. A recent systematic literature highlighted several supply chain barriers that contribute to high stockouts of contraceptives in LMICs, particularly in the last mile of the system in rural areas where service providers may be difficult to reach [7]. First, poor communication between different levels in the supply chain results in bottlenecks of commodities due to inadequate information on contraceptive inventory and consumption to inform supply chain decisions, including accurate forecasting of the type and quantity of contraceptives to procure [7,17,18]. Second, lack of reliable and well-maintained physical infrastructure, such as road networks and warehouses, as well as insufficient vehicles to navigate rural terrain, were noted as a major barrier to efficient transportation of supplies [7,19,20]. Finally, the absence of human resources dedicated to supply chain management leaves the responsibility of maintaining the supply chain to poorly trained staff and overburdened lower-level health workers, resulting in an erratic supply of contraceptives [7,17,20].

The contraceptive supply environment in the DRC is particularly challenging. As the largest country in SSA in terms of land mass, the vast size and poor transportation infrastructure pose massive barriers for distribution of health products [10]. Additionally, the government has neither the financial nor human capital to invest in a nationwide well-functioning supply chain. As a result, the current contraceptive supply chain is a weak piecemeal system dependent on the patchwork support of international donors [10]. In the last mile of the supply chain in the DRC, health products are traditionally distributed using a pull system in which lower-level health facilities order commodities as needed and are responsible for transporting the products from the HZ warehouse to the health facility. This system places a financial burden on health facilities that are operating with limited budgets [21]. Further, the pull system requires overworked health care providers to spend time on logistics rather than providing services to clients and is dependent on accurate and timely forecasting to predict what commodities will be needed, which can be challenging in low-resource areas where logistics training is limited [13,15,22]. The 2018 Service Provision Assessment (SPA) indicated that only half of the health facilities in the DRC that provide FP services had contraceptive methods in stock and available for clients on the day of the survey [23], indicating the supply chain is failing and Congolese women's access to contraception is threatened [24,25].

A relatively new supply chain model, termed the Informed Push Distribution Model (IPM), has shown promising results for strengthening the last mile of the contraceptive supply chain

in several other countries in SSA [20,26]. The first version of the IPM was introduced to the international FP field in Zimbabwe, where dedicated delivery truck teams were used to "top-up" health facilities' contraceptive stocks on a fixed schedule [26]. In the next version of the model, implemented in Senegal, a dedicated logistics manager was added to the system to improve stock reporting and help deliver contraceptives [20]. Following the success of the Senegal IPM, which demonstrated significant reductions in stockouts and improved communication throughout the logistics system [20,27], versions of the IPM have been adapted to a handful of other contexts, including the Direct Deliver and Information Capture system in Nigeria [28,29], the Dedicated Logistics Systems in Mozambique [30], and the IPM in Togo [31], with largely positive results. However, the model is highly influenced by the geography of each location; regional road infrastructure can affect the mode of commodity transportation and delivery time, and weather patterns (rainy or dry seasons) can make travel especially difficult in rural areas [28,29]. In addition, differences in resources, such as financial support and internet access, and varying levels of political support for FP require adapting implementation strategies of the IPM to each new context. Therefore, given the variability in environments and the strong interest in strengthening FP supply chains across SSA, it is important to test the IPM in a variety of contexts and adapt the model as needed.

The objective of this study was to demonstrate the feasibility and acceptability of implementing an adapted version of the IPM in the South Kivu province of the DRC to strengthen the last mile of the contraceptive supply chain. Located in the Eastern region of the country, South Kivu is characterized as a largely rural province with significantly weak road infrastructure, complex geographic challenges, and limited internet access. The province also faces high security risks from ongoing internal conflicts, as well as the recent and prolonged 2018 outbreak of Ebola in the region. The project began by piloting the model in one health zone in South Kivu, with the goal of using the lessons learned to further adapt and scale-up the model to more areas in the province or country.

## Methods

### Intervention

The pilot IPM was adapted to the DRC context to improve delivery in the last mile of the contraceptive supply chain in HZs which typically operated using a pull system that required the head nurse at each health facility to manage stock reporting, submit in-person reports and order requisitions to HZ management, and coordinate transportation of the requested commodities from the central HZ pharmacy to the health facility. The adapted pilot model, termed the Community-IPM (C-IPM), consisted of three changes to the supply chain: organization of the most efficient transportation routes, involvement of community members in the supply chain management, and bundling of contraceptives with other essential medications for delivery.

To adapt the C-IPM to the geographic environment, first the existing infrastructure and spatial distribution of health facilities was assessed to design optimal transportation routes from the HZ warehouse to facilities, resulting in five transportation routes, termed distribution axes in the model (Fig 1). Next, each axis in the model was assigned a C-IPM focal point by the existing HZ management structure, the *Comité de Développement Sanitaire* (Sanitary Development Committee), or CODESA, which is a community-led committee that manages the primary health care facilities in each HZ and leads health development in their area [32]. Each HZ in South Kivu is managed by a CODESA that consists of community health workers, known locally as *Relais Communautaires* (RECOs), who were elected to the committee, as described in detail elsewhere [32]. The C-IPM focal points were appointed by the CODESA in

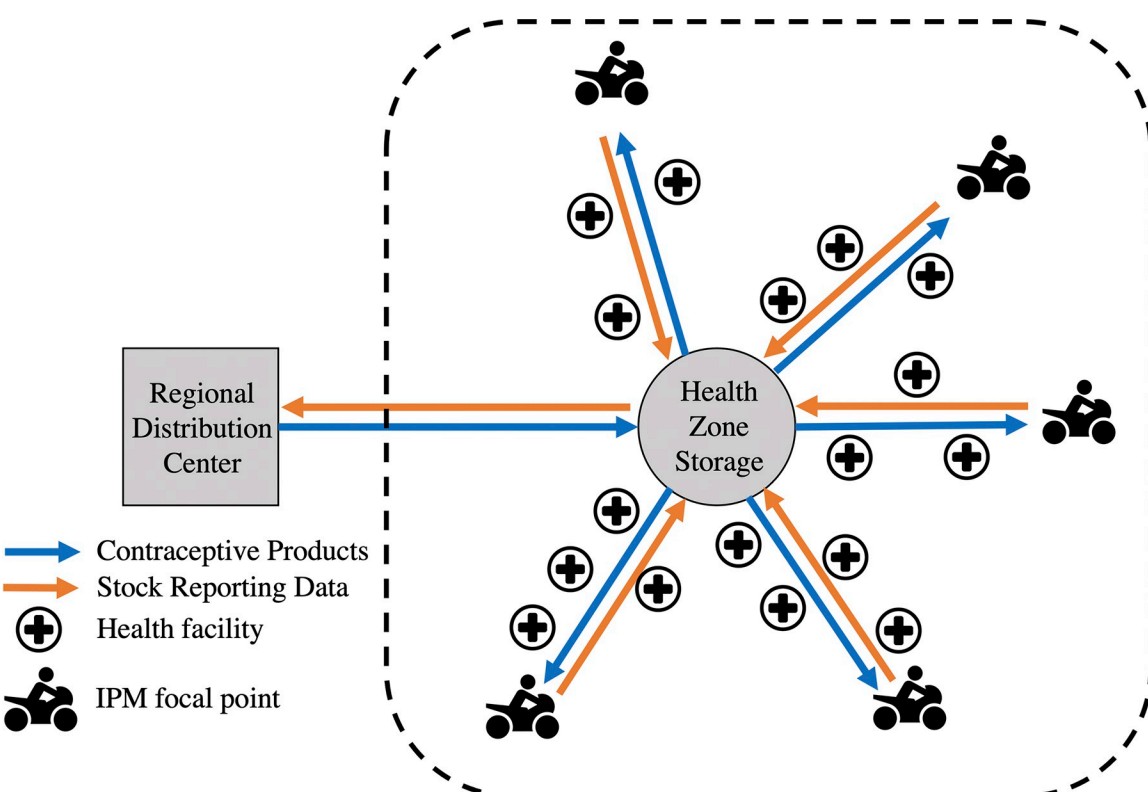

**Fig 1. The informed push distribution model (IPM) implemented in the last mile of the contraceptive supply chain of the DRC.**
IPM focal points were each assigned a distribution axis to deliver contraceptive products from the central HZ pharmacy to the health facilities along the axis.

Katana from within their list of committee members and were required to be able to read and write in the local language, have a good reputation in the community, and be willing to volunteer for the position. The decision to involve community members as logistic focal points in the pilot C-IPM was aimed at adapting the model to a low-resource setting to strengthen sustainability of the model by encouraging local buy-in to the project. The C-IPM focal points were not paid directly for their involvement in the intervention as it was considered a volunteer position; however, meals were served at all training sessions and transportation costs were reimbursed.

The C-IPM focal points had two main responsibilities. First, the focal points were responsible for collecting the monthly stock reports and order requisitions from the head nurse at each health facility along their assigned distribution axis and delivering these documents to the HZ pharmacy. Due to limited internet access in the HZ, stock reports and orders were prepared on paper by the head nurse. Each focal point set a monthly reporting date within the first 10 days of the month, in agreement with the HZ management. On the reporting date, the focal point would begin at the furthest health facility along their axis and drive up the route collecting stock reports and orders until reaching the HZ pharmacy. The focal point would then submit the documents to the HZ pharmacist who reviewed the monthly stock reports and order requisitions for accuracy and packaged the requested contraceptives into bundles for each distribution axis. If the HZ pharmacists noted issues in the reports, these were discussed with the head nurse of the facility at the monthly Monitoring Meeting, which is an institutionalized component of the health system in Katana where health facility staff and HZ management

gather to review and validate all data related to the health system. The second responsibility for the C-IPM focal points was then to deliver the requested FP commodities back down their distribution axis. To accomplish this task, the focal point would hire a local private motor taxi to pick up the contraceptive bundle and drive down the distribution axis to resupply the health facilities along the way.

Initially, only contraceptive commodities were distributed in the Katana HZ using the C-IPM. There were nine contraceptive methods included in the IPM, male condoms, female condoms, oral pills (Microgynon and Microlut), injectables (Depo-Provera), IUD, implants (Jadelle and Implanon NXT) and emergency contraceptive pills. However, four months after the pilot study began, the HZ management in Katana decided to expand the C-IPM to include essential medications for the treatment of malaria, tuberculosis, and HIV to maximize transportation efficiency and minimize overall delivery costs in the HZ. Under this change, the HZ pharmacists then included the expanded list of essential medications in the contraceptive bundles to be delivered as one package to the health facilities.

## Study design

The pilot C-IPM was tested in the Katana HZ of the South Kivu province of the DRC from October 2017 to October 2018. Katana is a rural HZ, approximately 45 km from the nearest city of Bukavu, which contains 20 health facilities that serve a population of 236,986. The local geography is mountainous, and the road infrastructure is unpaved. The area is inhabited mainly by the Bashi tribe and residents are predominately Christian. The C-IPM was implemented in all 20 health facilities in Katana in collaboration with the HZ management.

## Data sources

There were three main sources of data for this analysis. First, baseline and endline surveys of the health facilities in Katana were conducted in-person by local staff in October 2017 and 2018. The surveys included information on contraceptive availability, which was verified by observation by the local researcher, and stockouts, including the occurrence and duration of stockouts by method as reported in the health facility register. Second, monthly stock reports that contained information on daily commodity consumption, remaining stock levels, commodities received, and expired products were collected from health facilities by the C-IPM focal points throughout the study period and submitted to the HZ pharmacist. Following the monthly Monitoring Meeting, as described above, information from the stock reports were uploaded to the Logistic Management Information System (LMIS) in the DRC, along with measures for timeliness and completeness of the stock reports. For this analysis, data regarding the monthly stock reports was downloaded directly from the LMIS. Finally, the C-IPM focal points submitted cost reports to the health zone management for the transportation cost of delivering the commodity bundles each month.

## Key measures & analysis

First, *availability of contraception* was defined as a binary variable coded to 1 if there were no stockouts of any of the nine contraceptive methods included in this study at the health facility on the day of the survey. A stockout was defined as one or more methods of contraception unavailable or expired on the day of the survey. Next, the *duration of contraceptive stockouts* and *delivery time* were continuous variables measured in days. The *transportation cost* of delivering the commodities from the HZ pharmacy to the health facilities was a continuous variable measured in USD. Finally, two binary variables were used to measure the quality of the monthly reports submitted by the C-IPM focal points. The *timeliness of reports* was coded to 0

if the report was submitted one or more days after the agreed upon deadline and 1 if the report was submitted to the HZ management by the deadline. The *completeness of reports* was coded to 0 if any information on the report was missing and 1 if all information was included in the report.

Changes in the key measures between baseline and endline facility surveys were compared using chi-squared tests. Data were analyzed using STATA 17.0 (Stata Corp, College Station, Texas, US). Given the small sample size of health facilities, significance was assessed at the 0.10 level.

## Ethics statement

Ethical approval for this study was obtained from the Institutional Review Board of Tulane University, which determined the pilot evaluation did not constitute human subjects research as defined by the Common Federal Rule and thus review was not required (Study Reference #: 2023–1533). Additionally, the Provincial Department of Health (DPS) in South Kivu provided permission for the IPM intervention to operate in Katana and responded verbally that approval from the local ethics committee was not necessary given that no patient data was involved in the study. Additional information regarding the ethical, cultural, and scientific considerations specific to inclusivity in global research is included in the (S1 Checklist).

## Pilot study results

Availability of contraceptives increased among the 20 health facilities in Katana during the study period (Table 1). At baseline, 65% of facilities had all nine contraceptive methods in stock, while at endline this proportion had increased to 75%. The proportion of reports submitted on time increased from 75% to 80%, while the completeness of reports slightly decreased from 85% to 80%. However, the change in these two indicators was not statistically significant between baseline and endline surveys. The mean delivery time of products was also similar between baseline (1.6 days) and endline (1.7 days). However, there was a sharp decline in the mean duration of stockouts over the study period, from 7–12 days during the first two months of the study to 0–1 days throughout final 10 months of the study (Fig 2).

Initially, the delivery costs of contraceptive commodities from the central HZ pharmacy to the health facilities declined by the third month of the project, from $20 US to $5 US or less. Then in the fourth month of the study, the HZ management requested that other essential medications for the treatment of malaria, tuberculosis and HIV be packaged together with the contraceptive supplies and delivered using the same motor taxi along the same transportation routes to maximize transportation efficiency and minimize total delivery costs in the HZ. After this change in the study design, the delivery cost for the other essential medications also declined from $45 US at the beginning of the study to $2 US by the end of the study period (Fig 3).

**Table 1. Change in key indicators between baseline and endline surveys.**

|  | Baseline | Endline | p-value |
|---|---|---|---|
|  | n = 20 | n = 20 |  |
| All methods available at survey (%) | 65.0 | 75.0 | 0.49 |
| Reports submitted on time (%) | 75.0 | 80.0 | 0.71 |
| Reports completed (%) | 85.0 | 80.0 | 0.68 |
| Delivery time in days (mean) | 1.6 | 1.7 | 0.39 |

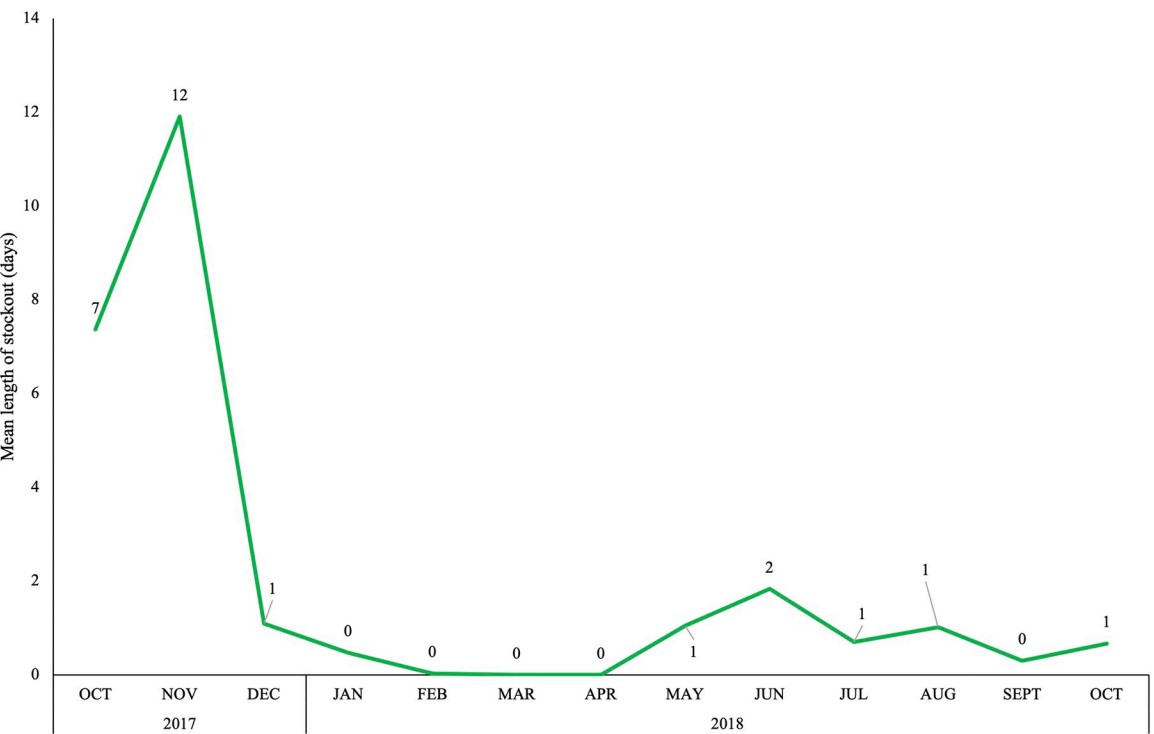

**Fig 2. Mean length of contraceptive stockouts at health facilities in the Katana health zone of South Kivu, DRC.**

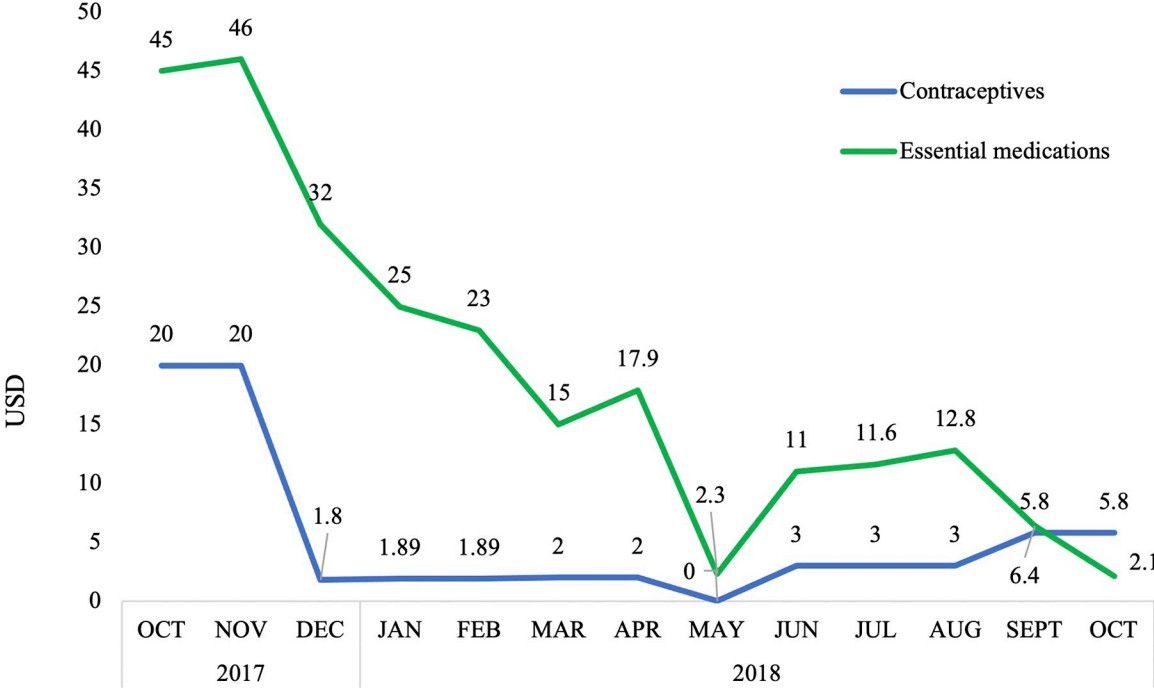

**Fig 3. Monthly transportation costs of delivery contraceptives and other essential medications through the adapted IPM to health facilities in the Katana health zone of South Kivu, DRC.**

## Discussion

This study is the first to implement an adapted version of informed push distribution model (IPM) in the DRC and contributes evidence to the feasibility of implementing the IPM to strengthen the last mile of the contraceptive supply chain in a challenging context. The flexible design of the C-IPM allows this intervention to be implemented in a wide variety of environments, which is essential in a country as large as the DRC with an array of geographies where one size will not fit all. By electing the C-IPM focal points from within the local community, this model is adaptable to a very low-resource setting which may improve sustainability beyond funding from international donors.

At baseline, 65.0% of health facilities in Katana had all nine contraceptive methods included in this study (male condoms, female condoms, oral pills (Microgynon and Microlut), injectables (Depo-Provera), IUD, implants (Jadelle and Implanon NXT) and emergency contraceptive pills) available on the day of the survey. Availability of FP in Katana was slightly higher than the national level (50%) reported in the 2017–18 SPA, which was surprising given the rural geography of the HZ [33]. However, several other international organizations conducted FP programs in this area prior to the C-IPM intervention, which may have increased general availability of FP services in Katana. While the timeliness of commodity reports from health facilities submitted to the HZ slightly increased during the study period (from 75.0% to 80.0%), the completeness of reports slightly decreased (from 85.0% to 80.0%). One possible explanation for the slight decline in completeness of reports may be that the head nurse did not finish the report by the reporting deadline, so when the C-IPM focal point came to the health facility to collect the documents the nurse had to submit an incomplete report. The average length of contraceptive stockouts initially increased, from 7 to 12 days. It is unclear why the mean stockout duration initially increased during the study period, this could be the result of challenges in procuring contraceptives further upstream in the supply chain. However, after the first two months of the study, the average duration of contraceptive stockouts decreased to less than two days throughout the remaining 10 months of the study, indicating the C-IPM was responsive to reported stockouts and able to quickly deliver the necessary commodities. Finally, the average transportation cost of delivering contraceptive supplies from the central HZ pharmacy to health facilities decreased by $14.20 US from the beginning to end of the study period, indicating the C-IPM was less expensive system to manage than the original pull system in Katana. After observing the decrease in transportation costs of contraceptives, the HZ authorities requested that other essential medications be included in the supply packages delivered along the C-IPM transportation routes. This request demonstrated that local health officials found the new supply chain model acceptable and suggests they were invested in sustaining the design beyond the study period.

The results of this study compare to findings from the pilot study that introduced the IPM in Senegal [20]. In both studies, stockouts of all contraceptive methods were quickly reduced in areas where the IPM was implemented and sustained throughout the study period [20]. Additionally, after the pilot study success in Senegal, regional health managers insisted on adapting the IPM to distribute other essential medications, similar to the request made by HZ authorities in the DRC study [20]. Although this addition to the model was not evaluated for cost-effectiveness in Senegal, results from this DRC pilot study show that bundling contraceptives with other essential medications streamlined the delivery process and lowered overall transportation costs in the health zone. A key difference between the Senegal IPM and the C-IPM in the DRC is that the Senegal model had strong support from the national government and was eventually scaled-up to the national level [20,27]. Although the DRC C-IPM gained strong support from the HZ authorities and the government in South Kivu, buy-in from government leaders in other provinces or national support will be necessary to scale-up the intervention.

## Limitations

There are some limitations to this study. First, there may have been other factors influencing the contraceptive supply chain in Katana during the pilot study period. A nurses' strike took place between May and August of 2018, during which stock reports were not submitted on time and supplies were not delivered to facilities resulting in stockouts of multiple contraceptive methods. Additionally, nurses at the health facilities may have been more thorough in their contraceptive stock management during the study period if they were aware their facility was being monitored by the project staff and HZ officials which may have contributed to the decrease in stockout duration. Further, activities from other FP organizations working in the area may have brought an influx of contraceptives at different times. Given the difficulty in coordination between FP projects in the area, which became apparent during the pilot study period, a contraceptive logistic group was added to the local Permanent Multisectoral Technical Committee (CTMP), a group of key stakeholders which guides FP initiatives in the provinces. The pilot C-IPM was only implemented in one rural HZ and the study design is not sufficient to determine causality between the model and changes in contraceptive availability or changes in use of contraception among women living in Katana. Thus, it will be important to rigorously evaluate the model using an experimental study design to determine whether the model significantly reduces stockouts and improves contraceptive availability compared to the existing pull supply chain system in the DRC.

## Lessons learned

A critical component to adapting and implementing the pilot C-IPM was the involvement of community members in the program. Working within the existing HZ management structure and involving community members as focal points in the supply chain enhanced community buy-in for the intervention quickly and may improve sustainability of the model beyond international donor funding. Additionally, bundling contraceptives with other essential medications streamlined the delivery process and lowered transportation costs for these medications in the HZ, which further strengthened health officials' support for the model. Finally, for the supply chain to function properly, it is essential that stock reports and contraceptive order requests are complete and turned in on time so the contraceptive needs for the HZ and province can be accurately forecasted and procured. Future IPM designs may consider implementing an incentive system to ensure prompt reporting and encourage increased ownership of the system by focal points.

## Future directions

After the C-IPM pilot study concluded, there was interest from health officials in South Kivu to expand the intervention to other rural HZs in the province. However, there is a large amount of variation in the HZs in South Kivu and some cover a very large geographic area with widely dispersed health facilities. Thus, the exact design of the C-IPM implemented in Katana would not be able to deliver contraceptive commodities to all health facilities in large HZs in only one day. Therefore, to adjust for the varying geography and size of HZs, the IPM model was further revised as spoke and hub model and renamed the *Comité Axes de Distribution* (CAD). In this revised model, a HZ is first divided into main distribution axes as done in the C-IPM, then along each distribution axis a health facility located near the middle of the transportation route is identified as a "hub" and transport branches are organized like "spokes" around the facility (Fig 4).

Focal points in the CAD model are elected by the local CODESA from within their list of community health worker members, similar to the C-IPM focal points, however, the CAD

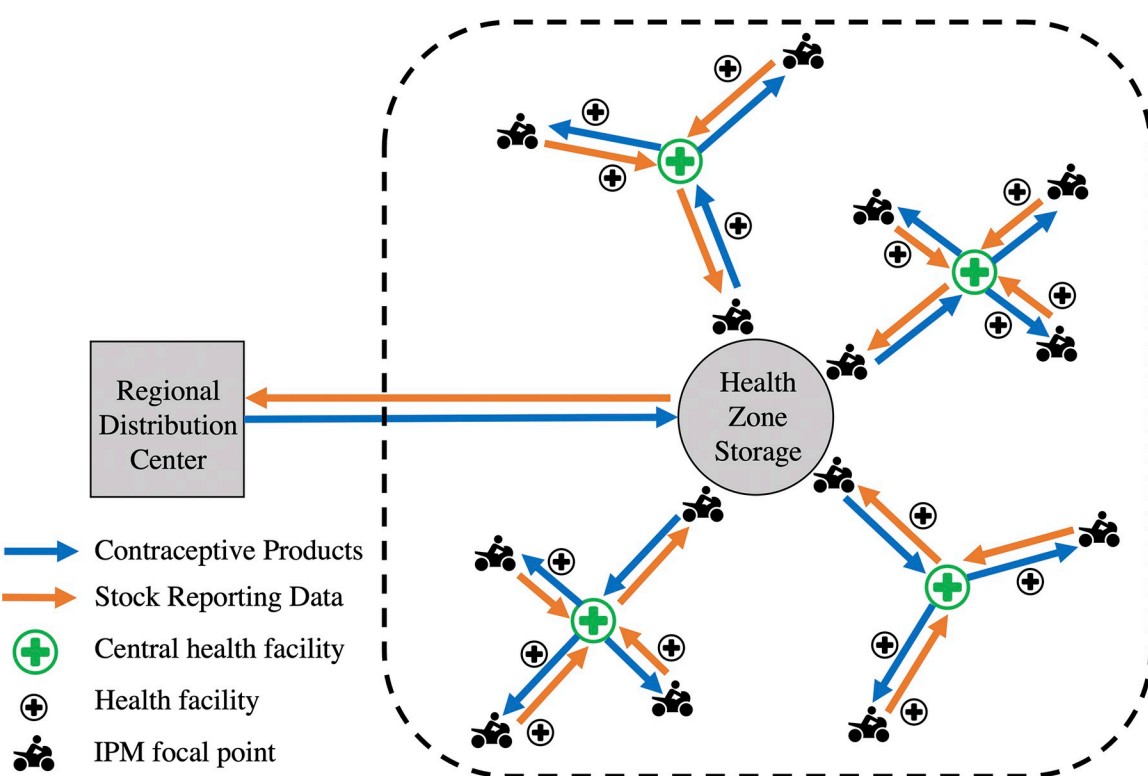

**Fig 4. Revised version of the adapted IPM, termed CAD, in the South Kivu province of the DRC.** Each distribution axis is divided into additional transport branches to efficiently cover a large geographic area in a short amount of time.

focal points are divided into two groups. The first group of CAD focal points are responsible for managing the transportation spokes around the health facility hub. The CAD focal points collect monthly stock reports and order requisitions from the health facilities along their assigned spoke and deliver these documents to the health facility hub. Once all documents from each spoke have been collected at the health facility hub, the second group of CAD focal points are responsible for picking up the stock reports and order requisitions from their assigned hub and delivering these to the HZ pharmacy. After the HZ pharmacist reviews the documents, the second group of CAD focal points picks up the requested commodities and delivers these back to their assigned health facility hub. Then, the first group of CAD focal points pick up the commodity bundle for their spoke and delivers the products to the health facilities along their distribution route.

The goal of the CAD is to efficiently cover a large geographic area in a short amount of time to keep the flow of contraceptive products moving on a reliable and timely schedule. However, to date, the CAD model has not been evaluated. Therefore, future studies are needed to assess whether the CAD model is more effective at ensuring contraceptive availability in large rural areas compared to the existing supply chain system.

## Conclusions

Overall, given the large land area of the DRC and poor national infrastructure, building a strong supply chain is challenging, however, the pilot study results indicate it is feasible to adapt the IPM to a new location with complex geographical, political and socioeconomical influences. Implementing a supply chain intervention like the C-IPM may be particularly

beneficial for strengthening contraceptive security in contexts where health facilities are over-burdened and staff do not have time to submit stock reports to higher management, especially in hard-to-reach areas with limited internet connection that require the use of physical documents. Further, the C-IPM may be especially useful in areas with fluctuating levels of demand for FP. For example, in some regions of the DRC, community-based programs operate by conducting intermittent outreach events connected to health facilities [34]. Thus, the C-IPM may support health facilities to prepare for anticipated increases in commodity consumption due to community events or other demand generating activities, such as FP messaging campaigns. Finally, the CAD model, designed based on lessons learned from the C-IPM pilot study, can alleviate the burden on health facility staff in large rural areas who had to spend considerable travel time picking up the contraceptive products themselves. In sum, the flexibility of the IPM to adapt to varied contexts make this an encouraging health system strengthening intervention to improve contraceptive security through the last mile of the supply chain to ensure all women have reliable access to a range of contraceptive methods.

## Supporting information

**S1 Checklist. Inclusivity in global research checklist.**
(DOCX)

## Acknowledgments

The authors wish to thank Jane Bertrand who provided valuable guidance in development and editing of the manuscript. Furthermore, the authors would like to thank consultant Freddy Salumu for his involvement in the design of the study, the presentation of the study protocol to health authorities and the implementation and initial monitoring of the pilot and consultant Gilbert Mungungu Witakenge for concluding the pilot and contributing to the final report. The authors also thank consultants Eugene Barhukenge and Justin Rukika, for their contributions in collecting and compiling data from the HZs, and the executives of South-Kivu DPS, Dr Zozo Musafiri and Dr Socrate Cuma for their contribution in the review of the methodology and the validation of the preliminary results. A special thanks to all HZ managers and FOSA service providers who took part in this study and the heads of the partner organizations who encouraged us and without whom this study would not have seen the light of day: Dr Arsene Binanga (Tulane University), Dr Dotian Wanogo (UNFPA), Dr Seydou Ndiay (Cordaid DRC) and Ph. Franck Biayi (PNAM/ MoH DRC). The Authors thank the Ministry of Foreign Affairs of the Netherlands and the Netherlands embassies in Rwanda and DRC who enabled realization of this study by funding the Jeune S3 project through Cordaid Netherlands, and the 3C project through UNFPA DRC.

## Author Contributions

**Conceptualization:** Michel Yalaza, Felix Hitayezu.

**Data curation:** Felix Hitayezu.

**Formal analysis:** Katherine H. LaNasa, Michel Yalaza.

**Investigation:** Michel Yalaza, Felix Hitayezu.

**Methodology:** Michel Yalaza, Felix Hitayezu.

**Project administration:** Frank Roijmans.

**Supervision:** Frank Roijmans.

**Validation:** Frank Roijmans.

**Writing – original draft:** Katherine H. LaNasa.

**Writing – review & editing:** Katherine H. LaNasa, Michel Yalaza, Felix Hitayezu, Frank Roijmans.

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
