## [Decision Letter · Decision Letter 0]

22 Feb 2024

PGPH-D-23-01903

Adapting the informed push model to the last mile of the contraceptive supply chain in the Democratic Republic of Congo: Pilot study results

Dear Dr. LaNasa,

Thank you for submitting your manuscript to PLOS Global Public Health. After careful consideration, we feel that it has merit but does not fully meet PLOS Global Public Health’s publication criteria as it currently stands. Therefore, we invite you to submit a revised version of the manuscript that addresses the points raised during the review process.

We look forward to receiving your revised manuscript.

Kind regards,

Anushka Ataullahjan

Guest Editor

Journal Requirements:

2. Please send a completed 'Competing Interests' statement, including any COIs declared by your co-authors. If you have no competing interests to declare, please state "The authors have declared that no competing interests exist". Otherwise please declare all competing interests beginning with the statement "I have read the journal's policy and the authors of this manuscript have the following competing interests:"

Additional Editor Comments (if provided):

Reviewers raise several important points about the methods, please read closely reviewer 1 and 5's comments and revise accordingly.

Reviewers' comments:

Reviewer's Responses to Questions

**Comments to the Author**

1. Does this manuscript meet PLOS Global Public Health’s publication criteria? Is the manuscript technically sound, and do the data support the conclusions? The manuscript must describe methodologically and ethically rigorous research with conclusions that are appropriately drawn based on the data presented.

Reviewer #1: Yes

Reviewer #2: No

Reviewer #3: Yes

Reviewer #4: Yes

Reviewer #5: Partly

2. Has the statistical analysis been performed appropriately and rigorously?

Reviewer #1: N/A

Reviewer #2: Yes

Reviewer #3: Yes

Reviewer #4: Yes

Reviewer #5: Yes

3. Have the authors made all data underlying the findings in their manuscript fully available (please refer to the Data Availability Statement at the start of the manuscript PDF file)?

Reviewer #1: Yes

Reviewer #2: Yes

Reviewer #3: No

Reviewer #4: Yes

Reviewer #5: Yes

4. Is the manuscript presented in an intelligible fashion and written in standard English?

Reviewer #1: Yes

Reviewer #2: Yes

Reviewer #3: Yes

Reviewer #4: Yes

Reviewer #5: Yes

5. Review Comments to the Author

Reviewer #1: This analysis of the informed push model in the DRC contributes to the contraceptive security knowledge base in challenging contexts. The manuscript is, for the most part, well-written. I have included questions and comments on the manuscript.

I suggest that the authors expand the introduction of the manuscript, including more information related to the challenges of contraceptive security globally. I also suggest expanding the discussion section of the manuscript, and add a "limitations" section and a "conclusions" section. In addition, it may be more logical to move the "lessons learned" section to AFTER the general "discussions" section. It would be helpful, in your conclusions, to discuss applicability of the informed push model to other challenging contexts.

Reviewer #2: Dear Author,

1. The title and abstract of manuscript needs resynthesis as in its current form it is not very informative

2. The methodology needs to be rewritten with appropriate subtitles and omission of word "Pilot" is highly recommended from, methodology section.

3. Discussion needs to be elaborate including more references.

Reviewer #3: It's a nice work, however, I have some comments:

1. I would suggest to revise the title as " :A pilot study results"

2. Please make the discussion section more clearer for the reader. Actually, home message must be clear and understandable for readers.

3. Refences are not well cited and maintained any specific style as per PGH guidelines. So, revise them carefully.

Thanks.

Reviewer #4: This article is clearly written and provides useful programmatic information about a promising approach to improving the FP supply chain.

It's interesting that the SPA survey found a 50% stockout rate nationally, but this study found 65% of facilities in the pilot study had all methods, even though this study was conducted in a remote area that might be expected to be worse that the country overall. Authors may want to comment on that in the discussion.

Minor notes:

Line 163 – I think it should say “reports” instead of reported

Line 195 - CODESA acronym should be spelled out on first reference.

Reviewer #5: Thank you for giving me the opportunity to review this paper. I found the topic very interesting and the study does contribute to filling a gap in drug delivery in resource-poor settings.

METHODS

Pilot Intervention

- Add timing of when the intervention started.

- How were the focal points identified from the community?

- It’s a bit difficult to understand what drugs the focal points initially started supplying at the beginning of the study and tie that with what was said in other parts of the manuscript. Did the monthly commodity consumptions being described in this section include all drugs at the health facilities or only essential drugs or only family planning items? In the Pilot Implementation and assessment section, stock reports were collected by the focal points (which is understandable) but then in the results, you describe how health facility staff and HZ management requested to include other essential drugs to the supply list. If my understanding is correct, the contraceptives were meant to be added to the essential medicines list identified for supply. So were these essential drugs not part of the supply list at the beginning of the study? It will be beneficial if you clearly explain what information (in terms of stocks/supplies) was collected at the beginning of the study and how this list changed over time, referring to the HZ management request to add other essential drugs.

Pilot implementation and assessment.

- Line 50: Please specify what you considered to be the key pilot indicators for the study.

- How did you define and assess timeliness of stock reports?

- Who did the focal points submit their reports to?

- I imagine the focal points were not necessarily medical professionals or pharmacists. So were there instances where mistakes in stock supplies occurred? And how were this supply mistakes dealt with if these did occur?

Ethics statement

- I am curious to know if attempts to obtain ethics approval from the national DRC ethics committee (National Committee of Health Ethics/ Comité National d’Éthique de la Santé) was sought for this study. And if not, why? Or were you already given a waiver by the DRC ethics committee?

Pilot results

- Line 163: change ‘reported’ to ‘reports’.

- For the reports described in the results section, I propose to specify what reports you are referring to; would that be the reports of monthly commodity consumption that the focal person collects from the nurse or is it the monthly reports that the focal person submits to track the duration of stockouts?

- How was completeness of the reports measured? What parameters were used? I suggest this explanation be added to the methods section.

- And what could have explained the decrease in completeness of reports from baseline to endline?

- Line 164: What was the definition of ‘on time’ for the study? i.e. was this exact to the pre-agreed date? What happened if the report was not on time? Could the focal person come back at a later date if the report you are referring to here is the nurse’s monthly commodity consumption? All this could be explained in the methods section.

- A brief description of how the delivery costs were calculated/determined in the methods section will be beneficial.

Lessons learned

- Line 194-196: The involvement of CODESA (please define as this is the first time it’s being used in the document) to appoint the focal persons from the community should have been first described in the methods section.

Revised intervention design

- At what time point in the study was the CAD implemented?

Discussion

- You may need to critically develop your discussion a bit further. For example, how did your results differ from other contexts that have implemented this IPM? And why? Or how are they similar if this is the case? Do you believe that the use of the focal points from the communities has markedly improved access to contraceptives (and other essential drugs) compared to other contexts? Did the improvements in the availability of the contraceptives at the health facilities translate to a higher uptake of these products by women? Is the decrease in time of stock-outs a reflection of the success of the IPM (and maybe improved uptake of the contraceptives) or was there the possibility that nurses were now being more conscientious in their stock management knowing that there is the added scrutiny of the study?

- What could explain the sharp increase in the mean length of stock-out days from October to November 2017?

6. PLOS authors have the option to publish the peer review history of their article (what does this mean?). If published, this will include your full peer review and any attached files.

**Do you want your identity to be public for this peer review?** For information about this choice, including consent withdrawal, please see our Privacy Policy.

Reviewer #1: **Yes: **Laura Miniea Hoemeke

Reviewer #2: No

Reviewer #3: **Yes: **Md. Shahjalal

Reviewer #4: No

Reviewer #5: **Yes: **Grace Mambula

---

## [Decision Letter · Decision Letter 1]

5 Jul 2024

Adapting the informed push model to the last mile of the contraceptive supply chain in South Kivu in the Democratic Republic of Congo

PGPH-D-23-01903R1

Dear Ms LaNasa,

We are pleased to inform you that your manuscript 'Adapting the informed push model to the last mile of the contraceptive supply chain in South Kivu in the Democratic Republic of Congo' has been provisionally accepted for publication in PLOS Global Public Health.

Best regards,

Julia Robinson

Executive Editor

Reviewer Comments (if any, and for reference):

Reviewer's Responses to Questions

**Comments to the Author**

1. If the authors have adequately addressed your comments raised in a previous round of review and you feel that this manuscript is now acceptable for publication, you may indicate that here to bypass the “Comments to the Author” section, enter your conflict of interest statement in the “Confidential to Editor” section, and submit your "Accept" recommendation.

Reviewer #1: All comments have been addressed

Reviewer #2: All comments have been addressed

Reviewer #4: All comments have been addressed

Reviewer #5: All comments have been addressed

2. Does this manuscript meet PLOS Global Public Health’s publication criteria? Is the manuscript technically sound, and do the data support the conclusions? The manuscript must describe methodologically and ethically rigorous research with conclusions that are appropriately drawn based on the data presented.

Reviewer #1: Yes

Reviewer #2: No

Reviewer #4: Yes

Reviewer #5: Yes

3. Has the statistical analysis been performed appropriately and rigorously?

Reviewer #1: Yes

Reviewer #2: No

Reviewer #4: Yes

Reviewer #5: Yes

4. Have the authors made all data underlying the findings in their manuscript fully available (please refer to the Data Availability Statement at the start of the manuscript PDF file)?

Reviewer #1: Yes

Reviewer #2: No

Reviewer #4: Yes

Reviewer #5: Yes

5. Is the manuscript presented in an intelligible fashion and written in standard English?

Reviewer #1: Yes

Reviewer #2: No

Reviewer #4: Yes

Reviewer #5: Yes

6. Review Comments to the Author

Reviewer #1: Thank you for addressing the concerns raised by reviewers on the previous version of this manuscript.

Reviewer #2: Dear Author,

The rebuttal against the queries are not satisfactory

Reviewer #4: No further comments.

Reviewer #5: (No Response)

7. PLOS authors have the option to publish the peer review history of their article (what does this mean?). If published, this will include your full peer review and any attached files.

**Do you want your identity to be public for this peer review?** For information about this choice, including consent withdrawal, please see our Privacy Policy.

Reviewer #1: **Yes: **Laura Hoemeke

Reviewer #2: No

Reviewer #4: No

Reviewer #5: **Yes: **Grace Mambula
